# Porous Carbon with Alumina Coating Nanolayer Derived from Biomass and the Enhanced Electrochemical Performance as Stable Anode Materials

**DOI:** 10.3390/molecules28062792

**Published:** 2023-03-20

**Authors:** Wasif ur Rehman, Haiming Huang, Muhammad Zain Yousaf, Farooq Aslam, Xueliang Wang, Awais Ghani

**Affiliations:** 1School of Mathematics, Physics and Optoelectronic Engineering, Hubei University of Automotive Technology, Shiyan 442002, China; 2School of Electrical and Information Engineering, Hubei University of Automotive Technology, Shiyan 442002, China; 3MOE Key Laboratory of Thermo-Fluid Science and Engineering, School of Energy and Power Engineering, Xi’an Jiaotong University, Xi’an 710049, China; 4Key Laboratory of Advanced Functional Materials and Mesoscopic Physics of Shaanxi Province, School of Physics, Xi’an Jiaotong University, Xi’an 710049, China

**Keywords:** biomass derived carbon, anode, alumina-rich coating, lithium-ion batteries, electrochemical performance

## Abstract

With the ever-increasing world population, the energy produced from green, environmentally friendly approaches is in high demand. In this work, we proposed a green and cost-effective strategy for synthesizing a porous carbon electrode decorated with alumina oxide (Al_2_O_3_) from cherry blossom leaves using the pyrolysis method followed by a sol-gel method. An Al_2_O_3_-coating nano-layer (4–6 nm) is formed on the porous carbon during the composition fabrication, which further adversely affects battery performance. The development of a simple rich-shell-structured C@Al_2_O_3_ nanocomposite anode is expected to achieve stable electrochemical performances as lithium storage. A significant contributing factor to enhanced performance is the structure of the rich-shell material, which greatly enhances conductivity and stabilizes the solid–electrolyte interface (SEI) film. In the battery test assembled with composite C@Al_2_O_3_ electrode, the specific capacity is 516.1 mAh g^−1^ at a current density of 0.1 A g^−1^ after 200 cycles. The average discharge capacity of carbon is 290 mAh g^−1^ at a current density of 1.0 A g^−1^. The present study proposes bioinspired porous carbon electrode materials for improving the performance of next-generation lithium-ion batteries.

## 1. Introduction

Lithium-ion batteries (LIBs) with high energy density and high charge/discharge capacity are in high demand for developing efficient energy storage systems [1,2]. Commercial anodes based on carbonaceous materials are used extensively in the present day as LIBs, because of their low cost, chemical stability, and superior conductivity. However, their low capacity, poor rate performance, and inherent safety issues need to be addressed [3,4,5]. To date, among anode materials, carbon offers significant potential because of its mechanical strength, abundant reserves, and low charging voltage [6,7]. Moreover, it is imperative to examine in detail how particle structure, including coatings and porous structures, impacts the electrochemical performance [8]. Further, continuous breakage and generation of the solid electrolyte interface (SEI) film leads to deteriorated battery performance [9].

Graphite has been used for lithium-ion batteries since 1991 when Sony released the first subsequently commercialized available batteries [10]. The discovery of high-performance carbonaceous materials and their application as battery anodes may be the key to the development of the next generation of LIBs. Recycling, reducing, and reusing waste in daily life is essential for converting it into green energy [11]. To accomplish this, more work needs to be done on exploring different biomass-derived carbon anodes [12,13,14]. These anodes have been synthesized in similar approaches (mainly via pyrolysis combined with acid or porogenic treatment), and display different performances as shown in Table 1. Typically, the performance of biomass anodes is judged based on their surface and internal structures, but this neglects how these differences affect their lithium-ion storage [14]. A commercial biomass-derived battery will not be possible without such studies to generate and optimize trends in the field [15,16]. In contrast, anode LIBs still suffer from low specific capacity and initial coulombic efficiency (ICE).

The abundance and low cost of biomass-derived carbon make it an ideal candidate for use in supercapacitors [30], LIBs [31], and sodium ion batteries SIBs [15] to address the challenges mentioned above. Additionally, porous nanostructured carbon biomass can serve as an interface between the lithium electrode and the electrolyte, allowing charge to transfer [32,33]. A graphite electrode with a theoretical value of 372 mAh g^−1^ was determined, which is lower to fulfill the current demands. Many approaches have been introduced to carbon-based materials to address these problems, including cellulose/Al_2_O_3_ [34], Al into novel Li_3_VO_4_@C [35], Si/Ni_3.4_Sn_4_/Al/C composites [36], Al/metal oxide composites [37], and Al doped carbon Li_4_Ti_5_O_12_ as lithium-ion batteries for anodes [38]. In addition, the expansion of silicon and the damage to the anode caused by electrolytes are relatively reduced. Various strategies are used such as chemical vapor deposition being used to synthesize composites with core-shell structures and varying carbon layers (2–30 layers) [39]. A layer of Al_2_O_3_ can improve the cycle stability of various Al/metal oxide composites by acting as an artificial SEI film [40,41]. A coating of Al_2_O_3_ also reduces lithium inventory loss but marginally reduces silicon active material loss. As a result, capacity decay is greatly reduced [42]. A facile and cost-effective G/Si composite with Al_2_O_3_ coatings was proposed by Zhu et al. [43]. Furthermore, the amorphous Al_2_O_3_ coating (thickness: 10–15 nm) enhances the electrochemical performance of the material and provides a stable artificial SEI layer.

Inspired by the above study, here we develop a simple two-step method, firstly deriving porous carbon activated via pyrolysis and secondly using a cost-effective sol-gel method. Biomass derived carbon showed a highly porous structure, which allowed the nanoparticles of alumina oxide. The final nanocomposite C@Al_2_O_3_ showed excellent electrochemical properties. In comparison with the modified pure carbon sample, the C@Al_2_O_3_ composite with the rich-shell structure is better in cycle and rate performance. The typical carbon sample has been delivered at the current rate of 0.1 mA g^−1^ with a discharge capacity of 290 mAh g^−1^. On the other hand, the C@Al_2_O_3_ nanocomposite provides an initial coulombic efficiency (ICE) of 65.9% and the anode reversible capacity is 516.1 mAh g^−1^ at a current rate of 0.1 A g^−1^ after 200 cycles. The compact Al_2_O_3_ layers on porous derived carbon, with resultant cycle stability, can contribute to ionic conductivity. Subsequently derived carbon enhances the conductivity of the composite. Due to the rich-shell alumina protection, the composite C@Al_2_O_3_ electrochemical performance is significantly improved. We believe a cost-effective sol-gel method would support new possibilities for lithium anode storage.

## 2. Results and Discussion

### 2.1. Structural Properties of Derived Porous Carbon and C@Al_2_O_3_

Figure 1 illustrates the process of the hierarchical structure of C@Al_2_O_3_ composition preparation. To activate biomass carbon, it should be cleaned of unwanted impurities and then further treated with KOH at 1000 °C. After pyrolysis and impurities are removed, carbon and Al_2_O_3_ are decorated with a low-cost sol-gel method. Due to the ultrasonic treatment process and the sol-gel and annealing process, the very small Al_2_O_3_ nanoparticles formed could be fast-wrapped in a carbon wall or adsorbed on its surface. Consequently, C@Al_2_O_3_ nanocomposite is fabricated in this experiment to be utilized as anode LIBs.

Carbon derived from cherry blossom waste is activated with KOH, and the corresponding morphology is shown in Figure 2a–c. Carbon materials with apparent high porosity are observed in Figure 2a,b, which is in the form of deep digs with meso/microporous sizes. Figure 2c shows the magnification of the deep digs with rough surface profile. In addition, both high-resolution images and high-resolution pore images indicate a two-dimensional layer. It can be concluded from the images above that the derived carbon is highly active and allows for encapsulation within alumina oxide nanoparticles. A porous structure, which has a higher conductivity, limits the volume expansion and electrochemical performance of anode LIBs. The SEM images displayed in Figure 2d–f show that the Al_2_O_3_ particles are well combined with carbon; these were intensively coated on the surface of porous carbon even after being sonicated for approximately half an hour. Al_2_O_3_ has a controlled and homogeneous growth mechanism, permitting solution deposition. As shown in Figure 2e,f, Al_2_O_3_ nanoparticles appear to aggregate on porous carbon-coated surfaces during annealing. Energy-dispersive X-ray spectroscopy (EDX) analysis was used to investigate scale morphology, composition, and particle distribution. The nanocomposite C@Al_2_O_3_ is shown in Figure 2g,h, and the elements Al, O, and C are detected. The EDX further shows the particles are distributed and well organized. In these images, the hybrid C@Al_2_O_3_ nanocomposite shows a compact coating and a free space structure, which promotes excellent electrode performance during cycling.

To better characterize the interface structure between the porous carbon and alumina particles, TEM was used to observe the microstructures of derived carbon porosity and C@Al_2_O_3_. Micro/mesopores and the rough area are two types of holes in carbon as shown in Figure 3a. Lithium-ion traffic is allowed during cycling because of this high porosity. Figure 3b, which is magnified from the yellow dotted box in Figure 3a, shows two different zones. Typically, deep holes are defined by micro/macro holes and carbon two-demission layers. Moreover, Figure 3c indicates that the carbon channels are highly interconnected due to roughness on the surface. A mesopore allows lithium ions to pass quickly through an electrolyte and electrode material, resulting in improved electrochemistry. Figure 3d for sample C@Al_2_O_3_ shows that Al_2_O_3_ is well decorated within the nanocomposite structure. In Figure 3e, Al_2_O_3_ nanoparticles are incorporated into the interstices of meso/micropores and covered in carbon walls. Moreover, C@Al_2_O_3_ hybrid composites that are highly packed with carbon protection are adsorbent, limiting the aggressive nature of the structure. In Figure 3f, a TEM image of a C@Al_2_O_3_ nanocomposite shows two distinct regions, with (f1) indicating the typical carbon profile and (f2) showing the Al_2_O_3_ pattern. Thus, carbon homogenously packed with Al_2_O_3_ could produce an exceptional performance as an anode LIB. The stable Al_2_O_3_ and carbon network was formed using this simple deposition method, which contributed to the overall electrochemical performance of the anode LIBs.

### 2.2. Material Structure Analysis

To investigate the phase evolution of carbon and the coated alumina particles in the biomass derived composite material, X-ray diffraction analyses (XRDs) were performed on the samples to determine the corresponding crystallinity. Figure 4a shows a spectrum of C@Al_2_O_3_ nanocomposite and pure carbon. A typical broad peak can be found in the nanocomposite showing the crystalline phase peaks at 26.01°, 41.6°, 44.20°, 52.2°, and 76.91°. The crystal planes of carbon are (002) and (100) which correspond to 26.01° and 46.01°, respectively. An excellent example of this is the conversion of Al(NO_3_)_3_ into Al_2_O_3_ without any byproducts occurring during annealing. According to TEM results, Al_2_O_3_ nanoparticles were formed out/inside the surface onto the carbon wall. According to Figure 4b, measuring the ratio of the disorder-induced band (D-band, ~1350 cm^−1^) to the graphitic band (G-band, ~1550 cm^−1^) in Raman spectra, D and G band are proportional to defect structure and ordered graphite structure separately. Further, the degree of amorphousness can be quantified and compared. It was found that porous carbon (I_D_/I_G_ = 0.9) > C@Al_2_O_3_ (I_D_/I_G_ = 0.83) is in order from most amorphous to least amorphous [44,45]. As biomass-derived carbons in general are amorphous, and the recrystallization of carbon requires high temperatures over 1000 °C for long periods, it is challenging for carbons to crystallize [46].

To evaluate the surface element distribution, X-ray photoelectron spectroscopy (XPS) was utilized to capture a high-resolution spectrum of C@Al_2_O_3_. Figure 4c show the survey spectra of the desired elements of C@Al_2_O_3_ nanocomposite such as Al 2p, O1s, and C 1s. Figure 4d presents the Al 2p peak due to the reaction between functional groups such as –NH_2_ and –OH groups, this 74.31 eV can be attributed to Al_2_O_3_ present on the surface of carbon layers [47,48]. From the fit of O 1s spectra displayed in Figure 4e, it can be found that there are three individual peaks centered at 531–532, 533–534, and 534–536 eV, which correspond to C=O quinone type groups (O-I), C–OH phenol groups/C–O–C ether groups (O-II), and COOH carboxylic groups (O-III) [49,50]. For C@Al_2_O_3_ composite samples, the carbon–oxygen functional groups can improve the wettability of carbon materials further to enhance the chemical reaction on the electrode/electrolyte surface for obtaining extra electrochemical performance energy storage applications [51]. Figure 4f shows the XPS spectrum of the C 1s, which can be divided into three major peaks of C=C/C–C (284.6 eV), C=O (286–287 eV), and COOH (289–290 eV) [50,52].

### 2.3. Electrochemical Performance of Porous Carbon and C@Al_2_O_3_

To evaluate the effect of the composite C@Al_2_O_3_ on the battery performance, the electrochemical tests are conducted on porous carbon and C@Al_2_O_3_ anode materials to determine their capacity to store charge. Throughout this study, weight was used to determine specific capacities. Figure 5a,b shows the CVs of the two electrodes at a scan rate of 0.1 mV s^−1^. There are three intense peaks in Figure 5a, the first cathodic scan, located at 0.2, 0.69, and 1.26 V, respectively. In carbon anodes, the second and third cycles appear at the same time and overlap nicely with each other [31]. It can be seen from Figure 5b that the irreversible reduction peak appears at the beginning of the lithiation process and disappears at the end of the process. Carbon can react with Li^+^ in this reaction, resulting in a solid electrolyte interface (SEI). The reduction of Li_x_Al_2_O_3_ is a broad cathodic peak at approximately 0.51 V. A typical peak in the anodic process of C@Al_2_O_3_ was observed between 1.2 and 2.0 V [53]. The facts above are consistent with previous studies [34]. A comparison of the charge/discharge profiles of pure carbon and C@Al_2_O_3_ hybrids is shown in Figure 5c,d for 0.1 A g^−1^. The C@Al_2_O_3_ sample had an initial coulombic efficiency of 65%, which resulted in 1150.9 mA h g^−1^ discharge capacity and 516.1 mA h g^−1^ charge capacity, and the electrodes continued to provide the same specific capacity after 200 cycles, achieving an average efficiency of 99.5%. The nanocomposite C@Al_2_O_3_ indicates good capacity retention and as a result is a promising candidate as a LIB anode.

It is very important to investigate the long cycling life and stability of the electrode materials. The rate capability of C@Al_2_O_3_ was further investigated at different current densities ranging from 0.1 to 1.0 A g^−1^ (Figure 5d). Using the C@Al_2_O_3_ electrode, discharge capacities of 518.6, 372.4, 310.3, 380.5, 230.3, and 220.1 mAh g^−1^ are achieved at 0.1, 2.0, 0.2, 0.5, 0.7, and 1.0 A g^−1^, respectively. Due to the high rate measurements, the C@Al_2_O_3_ electrode was able to recover to 515 mA g^−1^ at 1.0 A g^−1^. In this respect, it confirms the outstanding rate capability of C@Al_2_O_3_ electrodes. By examining the EIS of pristine carbon, and C@Al_2_O_3_, we further investigated the electrochemical behavior of the composite materials. An equivalent circuit for EIS is shown in Figure 5e. As C@Al_2_O_3_ electrodes have smaller semicircles than the other samples, C@Al_2_O_3_ composites have lower charge transfer resistance than C@Al_2_O_3_ (420 Ω). The R_ct_ of carbon, however, is still lower than that of C@Al_2_O_3_, which is due to its higher carbon content. This proves indirectly the importance of the carbon layer. Figure 5f illustrates that, after cycling, the charge resistance of the C@Al_2_O_3_ composite is much lower than (230 Ω), which indicates enhanced transport kinetics for effective electron conductions and electrode reactions. By reducing charge transfer resistance, more electrons and Li^+^ are transferred efficiently, resulting in improved electrochemical performance.

As shown in Figure 5g, the C@Al_2_O_3_ electrode has excellent cyclic performance with a high current density (0.1 A g^−1^). The C@Al_2_O_3_ nanocomposite’s first charge/discharge cycle is 1110.5 mAh g^−1^ while after 200 cycles it surpasses both pure carbons (290.1 mAh g^−1^) and hybrid C@Al_2_O_3_ (516.1 mAh g^−1^). Furthermore, C@Al_2_O_3_ has a capacity of approximately 516 mAh g^−1^ at 0.1 mA g^−1^, as shown in Figure 5h schematic, which indicates that C@Al_2_O_3_ has high cycling properties at high rates of charge and discharge. At different current rates, a moderate mass loading of Al_2_O_3_ delivered a high performance for C@Al_2_O_3_. A porous carbon had two factors that offset this effect: (1) its layered structure provided a channel for Li ions to flow rapidly during charging and discharging, and (2) the limited surface prevented excess Al_2_O_3_ nanoparticles from dispersing.

Carbon nanoparticles are pulverized during charge/discharge cycles, resulting in continuous increases in capacitance. During pulverization, the Al_2_O_3_ nanoparticles were exposed to more active sites, allowing for enhanced lithium storage, and the Al_2_O_3_ nanoparticles were in contact with carbon-rich encapsulation, allowing charge transfer to occur more efficiently. As cycling proceeds, a steady-state SEI on the surface of the C@Al_2_O_3_ electrode material would also be advantageous. The free transportation of lithium ion provides a free gateway for electron channels; as a result the electrode C@Al_2_O_3_ delivers excellent overall electrochemical performance as anode lithium storage.

In Figure 6a, TEM images show that, after 200 cycles, the C@Al_2_O_3_ particles are still present. There is still an intact structure based on the stability of the structure. The TEM image shows no further damage. In Figure 6b, we see the crystalline profile of C@Al_2_O_3_ coated with carbon. Carbon and Al_2_O_3_ particles play an essential role in stabilizing the structure and shortening the Li-ion pathway. According to Figure 6c, this structure exhibits three distinct characteristics: a rich coating of Al_2_O_3_; derived carbon reducing its strength; and cracked areas caused by lithiation strains. A porous structure is shown in Figure 6d,e for Li-ion transportation. In addition to retaining its structure after 200 cycles, the anode exhibits excellent electrochemical performance. Figure 6f shows clearly in high-resolution TEM images that the structure is still retained and shows the coating of alumina on the carbon surface, which demonstrates the cause of high electrochemical performance. Moreover, in Figure 6g, the schematic illustration of cycling performance with the behavior of the Li^+^ ion transport indicates the gateway for free electron flow within the electrode. The porous structure can effectively retrain the anode SEI and volume changes during lithiation or delithiation. These anode LIBs with enhanced electrochemical performance were obtained through the construction of carbon with Al_2_O_3_-rich shells using a cost-effective sol-gel method.

## 3. Experimental Section

### 3.1. Activation of Porous Carbon

Activated carbon refers to a wide range of carbonized materials with a high degree of porosity and high surface area. Activating with potassium hydroxide in terms of surface area and efficiency shows better results than sodium hydroxide for various applications [54]. This experiment involved drying cherry blossom leaves in sunny conditions. We washed the dried leaves several times and then dried them for two days at 50 °C. The leaves were crushed with a rotter and then set aside to become a fine powder. Following impregnation with potassium hydroxide (KOH), the crushed leaves were treated with KOH at a ratio of 1:1, then transferred to a crucible plate and heated at 1000 °C for one hour at 15 °C min^−1^. To remove impurities such as K, Mg, and so on, the sample was washed for 30 min with diluted hydrochloric acid (HCl). As a final step after the HCl treatment, the product was washed with deionized water.

### 3.2. Preparation of C@Al_2_O_3_

A typical preparation procedure of C@Al_2_O_3_ composites was as follows: To modify the surface of the above derived active carbon, 1.0 g of derived carbon was dispersed in a 100 mL PVA aqueous solution (1.5 wt%) for 8 h, then the suspension was filtered, washed with distilled water several times to remove the residual PVA, and dried at 70 °C in vacuum for 10 h. Subsequently, 0.3 g PVA-modified active carbon was dispersed into aluminum nitrate Al(NO_3_)_3_ (0.1 g) in 40 mL of ethylene glycol and stirring continued for 48 h at room temperature. Then the ammonia solution with a concentration of 1.5 wt% added dropwise was mixed thoroughly for four hours in an oil bath at 100 °C. After agitation for 1 h, the insoluble black products were filtered, washed with distilled water and ethanol, then dried at 70 °C in a vacuum overnight. The materials were annealed at 650 °C for 2 h under argon flow to obtain C@Al_2_O_3_ nanoparticles.

### 3.3. Material Characterization

To characterize the active materials, the X-ray diffraction (XRD) spectra of the powder samples were measured using an X-ray diffractometer (Bruker D8 ADVANCE), Almelo, Holland with Cu Kα (λ = 0.15405 nm) radiation source and further analyzed using X’Pert HighScore Plus software 5.1. The sample morphology was tested via transmission electron microscopy (TEM, JEM F200) and scanning electron microscopy (SEM, FEG 250), Tokyo, Japan. The thermal stability and the carbon content of the active materials were investigated using thermogravimetric and differential scanning calorimetry (TG-DSC, Mettler-Toledo), Geneva, Switzerland. The crystal defects were evaluated via Raman spectrum using a Jobin LabRam high-resolution spectrometer (HORIBA, France, SAS), in a scan range of 500–4000 cm^−1^ with a laser source of 532 nm. A Thermo Fisher, Tokyo, Japan, ESCALAB Xi+ spectrometer was used to analyze the C@Al_2_O_3_ structure and determine its chemical state. By using X-ray photoelectron spectroscopy (XPS, AXIS ultra DLD), using an Al Kα as the X-ray source, we measured the chemical composition of the active materials.

### 3.4. Electrochemical Measurements

A coin-type cell was assembled to investigate the electrochemical performances of the active materials. We prepared the anode electrodes by casting a slurry, which is composed of 75% active material, 15% super-P carbon, and 10% poly (vinylidene fluoride) (PVDF). It was determined that the active materials on the positive electrodes weighed ~1.5 mg cm^−2^. The cells were assembled using the positive electrodes, Li foils, and Cellgard 2325 separators. An amount of 1 M LiPF_6_ was dissolved in dimethyl carbonate and ethyl carbonate under a volume ratio of 1:1. A LAND-CT2001A battery test system was used for the galvanostatic charge/discharge experiments. The electrochemical impedance spectroscopy (EIS) and cyclic voltammetry (CV) were conducted using an electrochemical workstation with Versatile Multichannel Potentiostat 2/Z.

## 4. Conclusions

The study proposed a two-step process of C@Al_2_O_3_ composite fabrication, i.e., first deriving porous carbon from waste cherry leaves via pyrolysis and then compacting the Al_2_O_3_ rich-shell coating via a low-cost sol-gel method. The fabrication of porous carbon and Al_2_O_3_ samples with a well-organized structure and uniform packing force has been successfully accomplished. A porous C@Al_2_O_3_ composite performed excellently as an anode for lithium-ion batteries. A biomass carbon electrode and C@Al_2_O_3_ electrode showed a highly reversible capacity of 516.1 mAh g^−1^ with a current density of 0.1 A g^−1^ after 200 cycles. It was found that the carbon is highly stable; for example, at a current density of 0.1 A g^−1^ it achieved a reversible capacity of 290.1 mAh g^−1^. In the study, a dense, rich alumina coating (about 10 nm) was found to be the most effective method to prevent SEI formation, and enhanced electronic and ionic conductivity. The performance of a porous structure made of C@Al_2_O_3_ is excellent, demonstrating the importance of using biological waste with multi-chemically tunable structures for energy storage. The composite C@ Al_2_O_3_ material with enhanced electrochemical performance for LIBs displays great potential for being utilized to improve the performance and reduce the cost of LIBs.

## Figures and Tables

**Figure 1 molecules-28-02792-f001:**
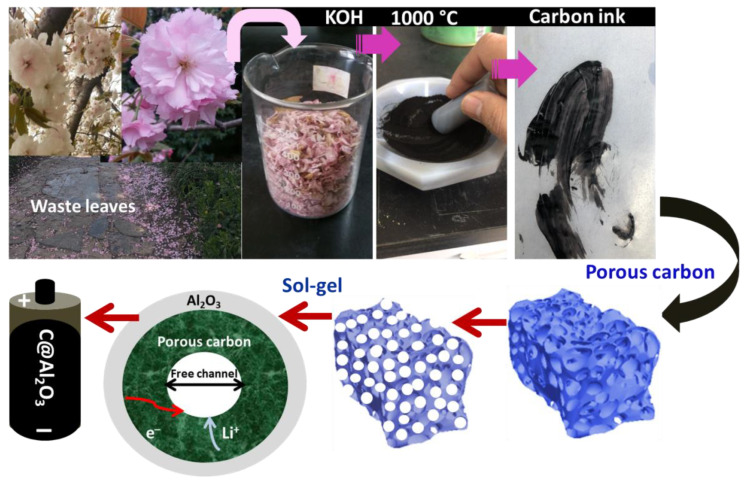
Schematic illustrations of carbon activation and C@Al_2_O_3_ composite preparation process.

**Figure 2 molecules-28-02792-f002:**
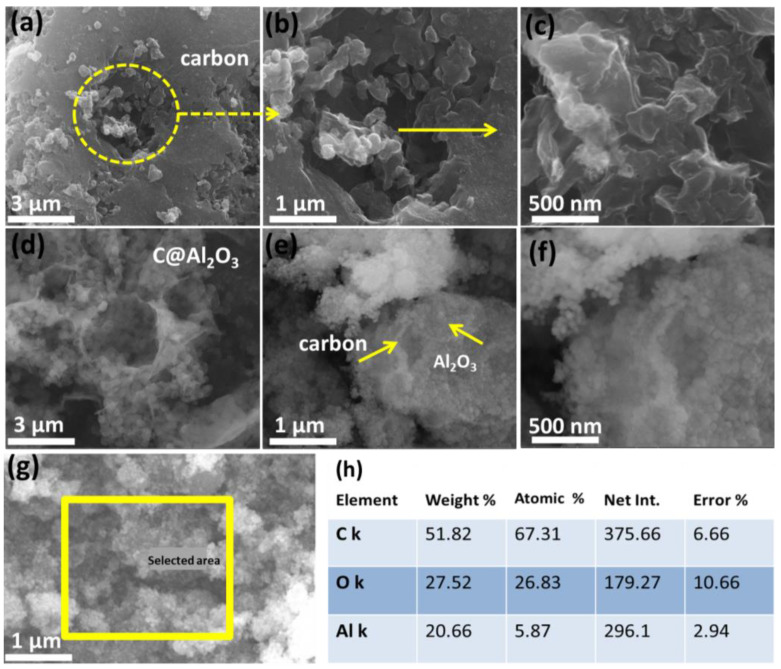
SEM images of cherry-blossom-derived porous carbon. (**a**,**b**) Porous morphology, (**c**) deep holes areas, (**d**–**f**) hybrid C@Al_2_O_3_ composition wrapped with a carbon wall, (**g**) EDS of C@Al_2_O_3_ selected area, (**h**) corresponding elements fraction of C, O, and Al in C@Al_2_O_3_.

**Figure 3 molecules-28-02792-f003:**
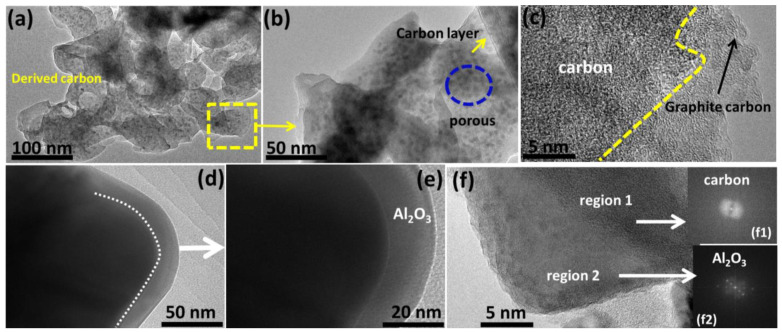
TEM images. (**a**,**b**) Derived carbon high porosity content area selection and (**c**) high-resolution TEM images of activated carbon different surface behavior. (**d**) TEM image of the C@Al_2_O_3_ composition distribution, (**e**) TEM image of the carbon coating and Al_2_O_3_ coating profile, and (**f**) two profiles, (**f1**) typical carbon and (**f2**) Al_2_O_3_ of C@Al_2_O_3_ nanocomposite.

**Figure 4 molecules-28-02792-f004:**
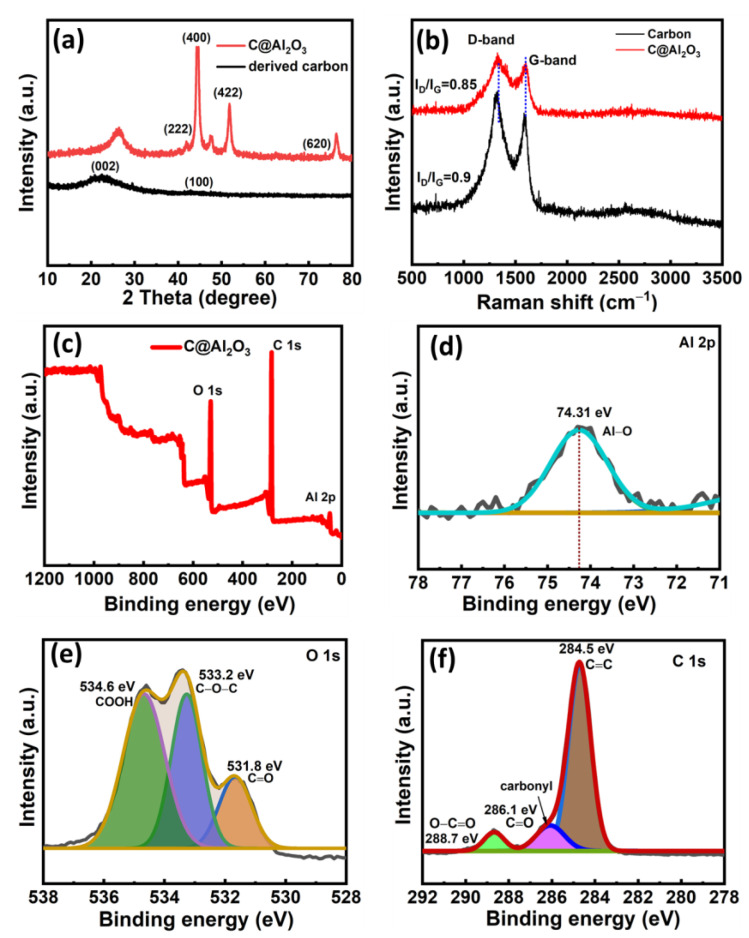
XRD pattern (**a**) of C@Al_2_O_3_, (**b**) Raman shift of derived carbon and C@Al_2_O_3_, (**c**) XPS survey pattern of C@Al_2_O_3_ sample, (**d**) Al 2p XPS plot, (**e**) XPS of oxygen (O 1s), (**f**) XPS pattern carbon C 1s.

**Figure 5 molecules-28-02792-f005:**
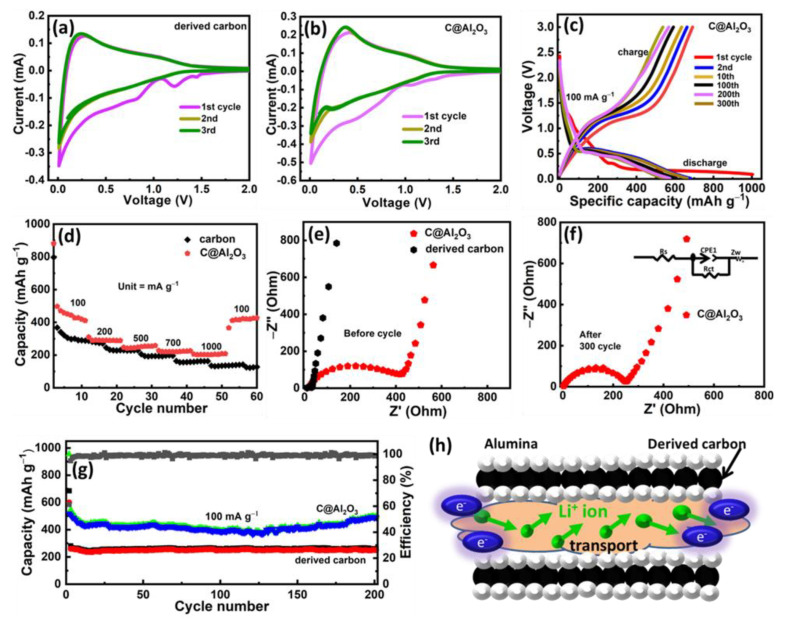
Electrochemical performances. (**a**,**b**) CVs of pure carbon and C@Al_2_O_3_, (**c**) charge/discharge profile of derived C@Al_2_O_3_, (**c**) charge-discharge vs. cyclic number of carbon C@Al_2_O_3_ sample at a current rate of 500 mA g**^−1^**, (**d**) rate capability of C and C@Al_2_O_3_ at various current densities of 100 to 1000 mA g**^−1^**, (**e**,**f**) EIS plot of two samples C, C@Al_2_O_3_, before and after cycles, (**g**) long cycling performance of the C@Al_2_O_3_ nanocomposite, and (**h**) scheme of Li-ion transportation cycling processes.

**Figure 6 molecules-28-02792-f006:**
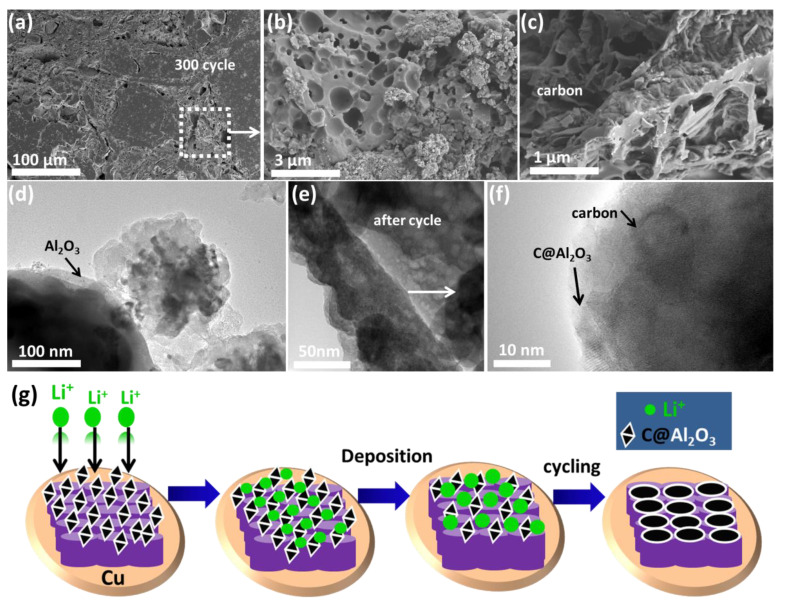
TEM images of C@Al_2_O_3_ electrodes (**a**) after 200 cycles and (**b**,**c**) structure stability, Al_2_O_3_ rich coating, crystalline profile and cracking regions, (**d**–**f**) Li-ion pathway during charge/discharge testing, (**g**) scheme of Li-ion transportation during cycling in charge/discharge process.

**Table 1 molecules-28-02792-t001:** Summary of the electrochemical performance of different carbon-based anodes in LIBs.

Sample	Specific Capacity (mAh g^−1^)	Cycles	Current (mA g^−1^)	References
Si/C	420.7	150	3C	2021 [17]
SiO_2_/C	530	100	500	2018 [18]
N/O doped carbon	307	500	1000	2022 [19]
Graphene- and rope-like nano carbons	355	1000	1000	2019 [20]
Microalgae-derived hollow carbon-MoS_2_	300	880	5	2019 [21]
ZIF-8 derived N-doped carbon/silicon composites	302	800	1	2021 [22]
Co0.85Se nanosheets/graphene	522.7	500	2	2018 [23]
Biomass-derived poly(furfuryl alcohol)-protected aluminum	400	25		2018 [24]
Selenium-doped carbon	450	580	0.5	2020 [25]
Tin–nanoparticle/carbon–nanofiber	430	200	0.1	2016 [26]
ZnS@N-doped carbon nanoplates	536	500	0.5	2021 [27]
Co/ZnO/nitrogen-doped carbon	400	50	0.2	2022 [28]
ZnFe_2_O_4_/C	579	100	0.1	2016 [29]
C@Al_2_O_3_	516	200	0.1	This work

## Data Availability

Not applicable.

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
