# Peer review of "Porous Carbon with Alumina Coating Nanolayer Derived from Biomass and the Enhanced Electrochemical Performance as Stable Anode Materials"

_molecules, 2023, doi:10.3390/molecules28062792_

Round 1

Reviewer 1 Report

Dear editor,

Thanks for offering this chance to review this work. The porous carbon-based electrode materials for LIBs derived from biomass is of great potential for the development of green electrical energy, which could help reduced the cost of the LIBs. While, to further improve the quality of this work, some issue should be well handled as follows,

1.     Why are the cherry blossom leaves selected for fabricating the porous carbon materials?

2.     Why is alumina nano-coating decorated on the porous carbon?

3.     The purpose of activation of the porous carbon in section 2.1 should be well illustrated to let readers know the necessity of this process.

4.     The scale bar and characters in figure 2g should be consistent with the others in figure 2.

5.     The ratio of peak intensity of D and G in Raman spectra should be calculated and displayed to quantitatively evaluate the crystal defect evolution in the porous carbon and C@alumina composite materials.

6.     To present more comprehensive background on this field, the following literature is recommended for authors to be compared and referenced in the literature review.

[1] Comparative simulation of thin-film and bulk-type all-solid-state batteries under adiabatic and isothermal conditions, Applied Thermal Engineering 223 (2023) 119957. doi.org/10.1016/j.applthermaleng.2022.119957

[2] Lithium ion transport in solid polymer electrolyte filled with alumina nanoparticles, Energy Adv., 2022, 1, 269. DOI: 10.1039/d2ya00025c.

[3] Analysis of Compression in Uniform and Non-Uniform GDL Microstructures on Water Transport, International Journal of Green Energy, 2021, 19 (13): 1389-1403.

[4] Open-Source CFD Elucidating Mechanism of 3D Pillar Electrode in Improving All-Solid-State Battery Performance, Adv. Sci. 2022, 2105454. DOI: 10.1002/advs.202105454

Author Response

Journal: Molecules

Manuscript Ref: molecules-2284611

Title: Porous carbon with alumina coating nanolayer derived from biomass and the enhanced electrochemical performance as stable anode materials

Author(s): Wasif ur Rehman, Huang Hai-Ming, Muhammad Tahir Zain Yousaf, Farooq Aslam, Xueliang Wang *, Awais Ghani

Responses to editor’s Comments:

We appreciate your detailed review and comments. We have revised the manuscript according to your suggestions and comments. These comments have been extremely valuable and helped us improve our manuscript. We made several revisions, which is considered to meet your expectations. The specific revisions made in the manuscript and answers to your questions are listed below. All the changes are marked in red color in the main manuscript and blue in this document.

Editor comments:

Reviewer’s Comment 1: Please check that all references are relevant to the contents of the manuscript.

Authors’ Reply: Thank you for bringing this to our attention. We checked all references and make sure their relevancy to the contents of the manuscript. All correction and modification are marked with red color in the revised manuscript.  

Reviewer’s Comment 2:  Any revisions to the manuscript should be marked up using the “Track Changes” function if you are using MS Word/LaTeX, such that any changes can be easily viewed by the editors and reviewers.

Authors’ Reply: Thank you so much for suggestions. We have followed your kind suggestions by using MS Word.

Reviewer’s Comment 3:  Please provide a cover letter to explain, point by point, the details of the revisions to the manuscript and your responses to the referees’ comments.

Authors’ Reply: Thank you for your kind guidelines. We provided a cover letter that explained the details of the revisions to the manuscript to responses referee’s comments.

Reviewer’s Comment 4:  If you found it impossible to address certain comments in the review reports, please include an explanation in your appeal.

Authors’ Reply: Thank you so much for providing general rules of the journal. We are respecting the reviewers’ valuables comments and we addressed all the comments, it helps us a lot to improve our manuscript.

Reviewer’s Comment 5:  The revised version will be sent to the editors and reviewers.

Authors’ Reply: Thank you so much for your time to polish our manuscript. We have modified all the mentioned mistakes and suggestions as per your guidelines and respected reviewers.

Reviewer#1

Thanks for offering this chance to review this work. The porous carbon-based electrode materials for LIBs derived from biomass are of great potential for the development of green electrical energy, which could help reduced the cost of the LIBs. While, to further improve the quality of this work, some issue should be well handled as follows,

Reviewer’s Comment 1:  Why are the cherry blossom leaves selected for fabricating the porous carbon materials?

Authors’ Reply: Thank you so much for such a valuable comments. In fact, biomass carbon sources are eco-friendly natural products compared to other carbon sources and have many advantages in preparing  carbon, including being inexpensive, easy to obtain, green, and abundant. In addition, the production of biomass carbon derived from natural biomass can convert low-value biomass waste into valuable and useful materials. The biomass containing heteroatoms is the best raw material for the preparation of porous carbon, in contrast to the porous carbon of man-made carbon sources that require the addition of external heteroatoms[1]. In recent years, various biomass has been used as carbon sources to prepare biomass derived carbon [2-10] (Figure 1). Frankly one day I was walk in school there were a lot of blossom leaves fall so I think might to be used as carbon source. Inspired from above beneficial research we used cherry blossom leaves as porous carbon source.

Figure.1, schematic illustration of biomass carbon sources

Reviewer’s Comment 2:  Why is alumina nano-coating decorated on the porous carbon?

Authors’ Reply: Thank you so much for point out this comment. In this study, in order to provide a favourable dispersibility and prevent harmful interface reaction for porous carbon in matrix, we propose a simple chemical precipitation to synthesize continuous alumina coating on the surface of porous carbon modified by PVP and urea functional group. The chemical precipitation process, the surface modification of carbon and the characterization of composites were discussed. Furthermore, other methods such as atomic layer deposition (ALD) and (CVD) are much expensive, therefore typical low-cost approach has been adopted.

Reviewer’s Comment 3:  The purpose of activation of the porous carbon in section 2.1 should be well illustrated to let readers know the necessity of this process.

Authors’ Reply: Thank you so much for highlighting this point. We agreed and explored clearly in main manuscript the processes of “activation of derived carbon”. Actually, activated carbon refers to a wide range of carbonised materials of high degree of porosity and high surface area. Activation with phosphoric acid is commonly used for lignocellulosic material and at lower temperatures. Zinc chloride generates more surface area than phosphoric acid but is used less due to environmental concerns. Potassium carbonate, in comparison with potassium hydroxide, produces higher yields and a higher surface area for the adsorption of large pollutant molecules such as dyes. Activating with potassium hydroxide in terms of surface area and efficiency shows better results than sodium hydroxide for various applications[11].

Reviewer’s Comment 4:  The scale bar and characters in figure 2g should be consistent with the others in figure 2.

Authors’ Reply: Thank you so much for your time. We have changed scale bar and characters in Figure 2g, in main manuscript and also as below.

Reviewer’s Comment 5:  The ratio of peak intensity of D and G in Raman spectra should be calculated and displayed to quantitatively evaluate the crystal defect evolution in the porous carbon and C@alumina composite materials.

Authors’ Reply: Thank you so much for your time. We are apologized to miss this point, right now added the modified Figure of Raman in main manuscript.

By measuring the ratio of the disorder-induced band (D-band, ~1350 cm–1) to the graphitic band (G-band, ~1550 cm–1) in Raman spectra, D and G band are proportional to defect structure and ordered graphite structure separately. Further the degree of amorphousness can be quantified and compared. It was found that porous carbon (ID/IG = 0.9) > C@Al2O3 (ID/IG = 0.83) in order from most amorphous to least amorphous [12, 13].

Reviewer’s Comment 6:  To present more comprehensive background on this field, the following literature is recommended for authors to be compared and referenced in the literature review.

Authors’ Reply: Thank you so much for valuable literature. We have added the literature and citied given research articles in main manuscript (ref# 24, 25, 8, and 10).

Reviwer#2

We appreciate your detailed review and comments. We have revised the manuscript according to your suggestions and comments. These comments have been extremely valuable and helped us improve our manuscript. We made several revisions, which we hope will meet your expectations. The specific revisions made to the manuscript and answers to your questions are listed below. All the changes are typed using red font in the main manuscript and blue in this document.

Reviewer’s Comment 1:  First of all, authors should carefully check the text of the manuscript and correct some typos (e.g., L. 45 "Recused waste"). They should also check their English.

Authors’ Reply: Thank you so much for your time. We have double checked these mistakes thoroughly and corrected in the revised manuscript.

Reviewer’s Comment 2:  All abbreviations must be decoded. For example, “SEI” is decoded only in Abstract.

Authors’ Reply: Thank you so much for highlighting these valuables comments which polished our manuscript. We have modified and changed all those abbreviation throughout the article. For example “ICE, Mg, and K etc. we have denoted in main manuscript red color.

Reviewer’s Comment 3:  It is not clear from the description what the novelty of the proposed synthesis method is. The method for the synthesis of porous biomass-based carbons is well described and investigated. The synthesis of Al2O3 is also well known. In addition, in the Experimental part there is no information how the authors combine the obtained porous carbon and aluminum oxide.

Authors’ Reply: Thank you so much for your time. We apologize for the miss of the full details of experimental process. The novel aspect is “other methods such as atomic layer deposition (ALD) and (CVD) are much expensive; therefore typical low-cost approach has been adopted. Secondly best of our knowledge first time biomass derived porous carbon can decorated with alumina layer. Furthermore, we “have modified and explained the experimental section of the C@Al2O3 composite in main manuscript.

A typical preparation procedure of C@Al2O3 composites was as follows: To modify the surface of above derived active carbon, 1.0 g of derived carbon was dispersed in a 100 ml PVA aqueous solution (1.5 wt%) for 8 h, then the suspension was filtered, washed with distilled water several times to remove the residual PVA, and dried at 70 °C in vacuum for 10h. Subsequently, 0.3 g PVA-modified active carbon was dispersed into aluminum nitrate Al(NO3)3 (0.1 g) in 40 mL of ethylene glycol and continuously stirring it for 48 hours at room temperature. Then the ammonia solution with a concentration of 1.5 wt% was added dropwise were mixed thoroughly for four hours in an oil bath at 100 °C. After agitated for 1 h, the insoluble black products were filtered, washed with distilled water and ethanol then dried at 70 °C in vacuum for overnight. The materials were annealed at 650 °C for 2 hours under argon flow to obtained C@Al2O3 nanoparticles.”

Reviewer’s Comment 4:  L. 93. This point also remains unclear. For what purpose the authors wash the alkali-soaked powder. Is the purpose to get rid of excess alkali, and for activation is used alkali, which had time to be adsorbed on the powder?

Authors’ Reply: Thank you so much for pointing our mistakes. Yes, the aim of washing the alkali-soaked powder is to get rid of the residual alkali. Furthermore, to make sure the purity of composite C@Al2O3, after annealing used the obtained product. We agreed with you that there was confusion in previous expression. We have revised the statement and provide complete details in the experimental section in the revised version.

Reviewer’s Comment 5:  L.186. The authors incorrectly indicated the names of the samples (should be reversed) in the XRD patterns. In addition, the text mentions the plane (001), the figure shows the correct one (100).

Authors’ Reply: Thank you so much for your time to point out our mistakes. We corrected in main manuscript now the text the plane is (100) which highlighted in red color.

Reviewer’s Comment 6:  L. 204. It would be more convenient for reading if the listing of peaks at the corresponding binding energies was done in sequence. Moreover, the attribution of these peaks raises questions. Also, the text does not correspond to the picture, where other information is given. As a rule, carbon materials contain different oxygen groups. How do the authors explain the presence of only two of them (and not three)?

Authors’ Reply: Thank you so much for mentioning this point. We agreed with you, now modified and followed your kind suggestions. We had corrected as well in the revised manuscript as follows.

For C 1s (Figure 4f), it is believed that the peaks at 284.5, and 286.1 and 288.7 eV correspond to graphitized carbon (C=C), carbonyls (C–O–C), and carboxyls (O–C=O), respectively whereas the components at 286.1 and 288.7 eV may be attributed to carbon bound to oxygen in surface species.

Reviewer’s Comment 7:  L. 207. Unfortunately, the analysis of the O1s region not only raises questions, but also looks incorrect. First, there are two peaks below the synthetic 532.8 eV peak. The authors should more carefully correlate their analysis with data from other publications on carbon materials.

Authors’ Reply: Thank you so much for your concern. We totally agreed with your opinion. We have double checked our XPS experiment and corrected for oxygen O 1s region.

The recorded Al 2p peaks are almost symmetric, whereas O 1s and C 1s core-level spectra can be deconvoluted into two and three individual peaks, respectively, as shown in Figures 4(e, f). The peak at 531.1 eV in Figure 4(e) may be attributed to Al–O bonds in the Al2O3 phase or C=O bonds in quinone groups. The other peak at 532.5 eV can be assigned to hydroxyl or carbonyl groups in quinines and anhydrides [14]. For C 1s (Figure 4f), it is believed that the peaks at 284.5, and 286.1 and 288.7 eV, corresponding to graphitized carbon (C=C), carbonyls (C–O–C), and carboxyls (O–C=O), respectively [15].

Figure 4, XRD, Raman, XPS patterns of porous carbon and C@Al2O3

Reviewer# 3

We appreciate your detailed review and comments. We have revised the manuscript according to your suggestions and comments. These comments have been extremely valuable and helped us improve our manuscript. We made several revisions, which we hope will meet your expectations. The specific revisions made to the manuscript and answers to your questions are listed below. All the changes are typed using red font in the main manuscript and blue in this document.

Reviewer’s Comment 1:  There are a lot of mistakes in grammar. Double-checking the English of the manuscript by an English native is required.

Authors’ Reply: Thank you so much for valuable comments. We are totally agreed and apologized for our mistakes. We double checked thoroughly through help of native person.

Reviewer’s Comment 2:  In line 18, “Porous carbon with rich Al2O3-coating nano-layer (4-6 nm) is formed during the lithiation” is written. Why is the porous carbon formed during the lithiation?

Authors’ Reply: Thank you so much for highlighting this mistake. The focus was the rich coating of alumina during fabrication not lithiation now we corrected in the revised manuscript “Al2O3-coating nano-layer (4-6 nm) is formed on the porous carbon during the composition fabrication, which further adversely affects battery performance”.

Reviewer’s Comment 3:  From line 76-88, it seems like the C@Al2O3 sample was firstly fabricated from cherry blossom leaves and then coated by Al2O3 precursor via sol-gel process. However, in the experimental section 2.2 (Preparation of C@Al2O3), the cherry blossom leaves derived carbon was not added. It is really confusing.

Authors’ Reply: Thank you so much for your time. We have added the missing statement and explained completely the experimental process as per your kind suggestion. We are highly respecting your valuable comments which help a lot to improve our manuscript. We have modified our experimental section in main manuscript as well as below.  

A typical preparation procedure of C@Al2O3 composites was as follows: To modify the surface of above derived active carbon, 1.0 g of derived carbon was dispersed in a 100 ml PVA aqueous solution (1.5 wt%) for 8 h, then the suspension was filtered, washed with distilled water several times to remove the residual PVA, and dried at 70 °C in vacuum for 10h. Subsequently, 0.3 g PVA-modified active carbon was dispersed into aluminum nitrate Al(NO3)3 (0.1 g) in 40 mL of ethylene glycol and continuously stirring it for 48 hours at room temperature. Then the ammonia solution with a concentration of 1.5 wt% was added dropwise were mixed thoroughly for four hours in an oil bath at 100 °C. After agitated for 1 h, the insoluble black products were filtered, washed with distilled water and ethanol then dried at 70 °C in vacuum for overnight. The materials were annealed at 650 °C for 2 hours under argon flow to obtained C@Al2O3 nanoparticles.

Reviewer’s Comment 4:  In the line 82, ICE is the abbreviation which has not been defined. In figure 2d-f, what is the point to show three images of one sample? Besides, it is hard to tell that the Al2O3 is uniformly coated on the carbon. The elemental mapping should be provided.

Authors’ Reply: Thank you so much for guidance. We have defined the abbreviation “initial coulombic efficiency (ICE) in the revised manuscript. The aim of these three images from one sample was to provide high-resolution SEM images to exhibits the selected area of porous carbon morphology. In Figure 2(a-c) the derived porous carbon images different resolution (like 3µm, 1µm and 500 nm). The distribution of alumina on the porous is not that uniform due to the aggregation induced by the relatively high fraction indicated in the corresponding Energy-dispersive X-ray spectroscopy (EDX). Consequently, the mapping is not offered here.

Reviewer’s Comment 5:  In the TEM, the graphitized carbon can be observed. However, in the XPS C 1S analysis, the C=C signal cannot be detected.

Authors’ Reply: Thank you so much for your time to improving our manuscript. We are apologized for the incorrect description, which has been revised in the updated manuscript. For the derived porous carbon from cheery blossom, TEM images confirm that a graphitic nature (e.g., graphite, graphene, carbon nanotubes etc.) will have a C 1s main peak, attributed to C=C, which can be used as a charge reference set to 284.5 eV. An average of values for graphite from references from the NIST database [16] is 284.46 eV with a standard deviation [17, 18]. We have changed the derived carbon C 1s Figure as below and in the revised manuscript.

Figure 5, XPS spectrum of C 1s element

References

[1] Z. Li, Q. Wang, Z. Zhou, S. Zhao, S. Zhong, L. Xu, Y. Gao, X. Cui, Green synthesis of carbon quantum dots from corn stalk shell by hydrothermal approach in near-critical water and applications in detecting and bioimaging, Microchemical Journal, 166 (2021) 106250.

[2] X. Feng, Y. Jiang, J. Zhao, M. Miao, S. Cao, J. Fang, L. Shi, Easy synthesis of photoluminescent N-doped carbon dots from winter melon for bio-imaging, RSC Advances, 5 (2015) 31250-31254.

[3] R. Bandi, B.R. Gangapuram, R. Dadigala, R. Eslavath, S.S. Singh, V. Guttena, Facile and green synthesis of fluorescent carbon dots from onion waste and their potential applications as sensor and multicolour imaging agents, RSC Advances, 6 (2016) 28633-28639.

[4] A. Sachdev, P. Gopinath, Green synthesis of multifunctional carbon dots from coriander leaves and their potential application as antioxidants, sensors and bioimaging agents, Analyst, 140 (2015) 4260-4269.

[5] S. Zhao, M. Lan, X. Zhu, H. Xue, T.W. Ng, X. Meng, C.S. Lee, P. Wang, W. Zhang, Green Synthesis of Bifunctional Fluorescent Carbon Dots from Garlic for Cellular Imaging and Free Radical Scavenging, ACS Appl Mater Interfaces, 7 (2015) 17054-17060.

[6] N. Wang, Y. Wang, T. Guo, T. Yang, M. Chen, J. Wang, Green preparation of carbon dots with papaya as carbon source for effective fluorescent sensing of Iron (III) and Escherichia coli, Biosensors and Bioelectronics, 85 (2016) 68-75.

[7] C. Wang, D. Sun, K. Zhuo, H. Zhang, J. Wang, Simple and green synthesis of nitrogen-, sulfur-, and phosphorus-co-doped carbon dots with tunable luminescence properties and sensing application, RSC Advances, 4 (2014) 54060-54065.

[8] B.S.B. Kasibabu, S.L. D'Souza, S. Jha, R.K. Singhal, H. Basu, S.K. Kailasa, One-step synthesis of fluorescent carbon dots for imaging bacterial and fungal cells, Analytical Methods, 7 (2015) 2373-2378.

[9] V.N. Mehta, S. Jha, S.K. Kailasa, One-pot green synthesis of carbon dots by using Saccharum officinarum juice for fluorescent imaging of bacteria (Escherichia coli) and yeast (Saccharomyces cerevisiae) cells, Materials Science and Engineering: C, 38 (2014) 20-27.

[10] W. Lu, X. Qin, S. Liu, G. Chang, Y. Zhang, Y. Luo, A.M. Asiri, A.O. Al-Youbi, X. Sun, Economical, green synthesis of fluorescent carbon nanoparticles and their use as probes for sensitive and selective detection of mercury(II) ions, Anal Chem, 84 (2012) 5351-5357.

[11] Z. Heidarinejad, M.H. Dehghani, M. Heidari, G. Javedan, I. Ali, M. Sillanpää, Methods for preparation and activation of activated carbon: a review, Environmental Chemistry Letters, 18 (2020) 393-415.

[12] C. Huang, Z. Feng, F. Pei, A. Fu, B. Qu, X. Chen, X. Fang, H. Kang, J. Cui, Understanding Protection Mechanisms of Graphene-Encapsulated Silicon Anodes with Operando Raman Spectroscopy, ACS Applied Materials & Interfaces, 12 (2020) 35532-35541.

[13] Y. Ma, A. Huang, Y. Li, H. Jiang, W. Zhang, L. Zhang, L. Li, S. Peng, Simple preparation of Si/N-doped carbon anodes from photovoltaic industry waste for lithium-ion batteries, Journal of Alloys and Compounds, 890 (2022) 161792.

[14] L. Lin, W. Lin, Y.X. Zhu, B.Y. Zhao, Y.C. Xie, G.Q. Jia, C. Li, Uniformly Carbon-Covered Alumina and Its Surface Characteristics, Langmuir, 21 (2005) 5040-5046.

[15] I.V. Plyuto, A.P. Shpak, J. Stoch, L.F. Sharanda, Y.V. Plyuto, I.V. Babich, M. Makkee, J.A. Moulijn, XPS characterisation of carbon-coated alumina support, Surface and Interface Analysis, 38 (2006) 917-921.

[16] M.C. Biesinger, Accessing the robustness of adventitious carbon for charge referencing (correction) purposes in XPS analysis: Insights from a multi-user facility data review, Applied Surface Science, 597 (2022) 153681.

[17] C.D. Wagner, A.V. Naumkin, A. Kraut-Vass, J.W. Allison, C.J. Powell, J.R.Jr. Rumble, NIST Standard Reference Database 20, Version 3.4 (web version) (http:/srdata.nist.gov/xps/) 2003

[18] D.J. Morgan, J. Carbon. Res. 7 (2021) 51.

Reviewer 2 Report

In the manuscript entitled "Porous carbon with alumina coating nanolayer derived from biomass and the enhanced electrochemical performance as stable anode materials", the authors have prepared C@Al2O3 composite material and tested it as anode material in the battery.

Unfortunately, I can recommend this manuscript for publication in the journal only major revision. The main observation is that the manuscript lacks novelty. Although the study is interesting, the scalability of cherry blossom leaves is questionable.

The manuscript contains some other shortcomings. Please find below my comments.

1)      First of all, authors should carefully check the text of the manuscript and correct some typos (e.g., L. 45 "Recused waste"). They should also check their English.

2)      All abbreviations must be decoded. For example, “SEI” is decoded only in Abstract.

3)      It is not clear from the description what the novelty of the proposed synthesis method is. The method for the synthesis of porous biomass-based carbons is well described and investigated. The synthesis of Al2O3 is also well known. In addition, in the Experimental part there is no information how the authors combine the obtained porous carbon and aluminum oxide.

4)      L. 93. This point also remains unclear. For what purpose the authors wash the alkali-soaked powder. Is the purpose to get rid of excess alkali, and for activation is used alkali, which had time to be adsorbed on the powder?

5)      L.186. The authors incorrectly indicated the names of the samples (should be reversed) in the XRD patterns. In addition, the text mentions the plane (001), the figure shows the correct one (100)

6)      L. 204. It would be more convenient for reading if the listing of peaks at the corresponding binding energies was done in sequence. Moreover, the attribution of these peaks raises questions. Also, the text does not correspond to the picture, where other information is given. As a rule, carbon materials contain different oxygen groups. How do the authors explain the presence of only two of them (and not three)?

7)      L. 207. Unfortunately, the analysis of the O1s region not only raises questions, but also looks incorrect. First, there are two peaks below the synthetic 532.8 eV peak. The authors should more carefully correlate their analysis with data from other publications on carbon materials.

Author Response

(The authors gave the same response as above.)

Reviewer 3 Report

The genesis of Wasif ur Rehman’s paper is to demonstrate a two-step synthetic route to prepare C@Al2O3 and its potential functional usage as electrodes material in the Li-ion battery. However, this similar strategy has been widely reported. Furthermore, the writing and description is not well-organized. Therefore, this article is not recommended for publication in Molecules.

The areas that is doubtable are listed below.

1. There are a lot of mistakes in grammar. Double-checking the English of the manuscript by an English native is required.

2. In line 18, “Porous carbon with rich Al2O3-coating nano-layer (4-6 nm) is formed during the lithiation” is written. Why is the porous carbon formed during the lithiation?

3. From line 76-88, it seems like the C@Al2O3 sample was firstly fabricated from cherry blossom leaves and then coated by Al2O3 precursor via sol-gel process. However, in the experimental section 2.2 (Preparation of C@Al2O3), the cherry blossom leaves derived carbon was not added. It is really confusing.

4. In the line 82, ICE is the abbreviation which has not been defined.

In figure 2d-f, what is the point to show three images of one sample? Besides, it is hard to tell that the Al2O3 is uniformly coated on the carbon. The elemental mapping should be provided.

5. In the TEM, the graphitized carbon can be observed. However, in the XPS C 1S analysis, the C=C signal cannot be detected.

To put my inputs together, the revised the manuscript is not acceptable. Some polish in writing could be made to make it clearer and crisper.

Author Response

(The authors gave the same response as above.)

Round 2

Reviewer 2 Report

The attitude toward the comments seems very careless. Please, find below new remarks. Take for example note 5, which deals with incorrect legends in XRD ("the names of the samples (should be reversed) in the XRD pattern"). Unfortunately, the authors have not corrected the pattern. Although it is obvious that the carbon material should give a smoother XRD pattern, while the composite gives a pattern with clear peaks related to the metallic phase. Apparently, the authors were in a hurry with these responses and did not take the time to understand what the reviewer was writing. This is disappointing.

 1)      It is generally agreed that the abbreviation transcription exists independently in the Abstract and in the main body of the article. Therefore, once again, I recommend that the transcription for the SEI abbreviation be placed not only in the Abstract, but also in the first mention in the manuscript. On the other hand, the authors have inserted unnecessary, in my opinion, transcripts. For example, the names of chemical elements.

2)  Still, it remains unclear why the authors so thoroughly wash the leaf powder from the alkali. In my opinion, the added phrase "to get rid of the residual KOH" is not quite correct, because one might get the impression that the authors completely remove the alkali and then the question arises as to why they used it at all. Also, if we are talking about a porous carbon material, it would be nice if the authors gave the nitrogen adsorption/desorption isotherm for the final composite material.

3) Unfortunately, the authors have not corrected the XRD pattern (Fig. 4a). It would also be proper to specify in the experimental part the parameters of the XPS study (cathode, power of the source, analysis features like a type of background, peak functions) and the wavelength for Raman.

4) L. 224. Repeated use of XPS transcripts. Unfortunately, the authors ignored the question as to why there are no carbonyl groups in the activated carbon material.

5) L. 229. “These results indicate that the derived carbon is derived from flower waste leaves” – what does it mean? How XPS results can tell that?

6) The interpretation of the XPS spectra is very poor. The authors give the spectrum of the O1s region, claiming that carbonyl groups are present there, but there is no such peak on the spectrum of the C1s region. In addition, in my opinion, the authors should be more thoroughly examined in the articles where carbon materials are studied using XPS. The point is that the authors incorrectly attributed the peaks on the O1s spectrum. The peak from carbonyl and carboxyl groups is in the region of lower binding energies compared to single C-O bonding.

Author Response

We appreciate your detailed review and comments. We have revised the manuscript according to your suggestions and comments. These comments have been extremely valuable and helped us improve our manuscript. We made several revisions, which we hope will meet your expectations. The specific revisions made to the manuscript and answers to your questions are listed below. All the changes are typed using red font in the main manuscript and blue in this document.

Reviewer’s Comment 1:  It is generally agreed that the abbreviation transcription exists independently in the Abstract and in the main body of the article. Therefore, once again, I recommend that the transcription for the SEI abbreviation be placed not only in the Abstract, but also in the first mention in the manuscript. On the other hand, the authors have inserted unnecessary, in my opinion, transcripts. For example, the names of chemical elements.

Authors’ Reply: Thank you so much for your time. We had mentioned the abbreviation of SEI in the manuscript accordingly. We also changed the names of chemical elements just used abbreviation as before.

Reviewer’s Comment 2:  Still, it remains unclear why the authors so thoroughly wash the leaf powder from the alkali. In my opinion, the added phrase "to get rid of the residual KOH" is not quite correct, because one might get the impression that the authors completely remove the alkali and then the question arises as to why they used it at all. Also, if we are talking about a porous carbon material, it would be nice if the authors gave the nitrogen adsorption/desorption isotherm for the final composite material.

Authors’ Reply: Thank you so much for highlighting this mistake again. We apologize for this careless expression. We fully agreed with your point and revised the experimental process in revised manuscript as follows,

“This experiment involved drying cherry blossom leaves in sunny conditions. We washed the dried leaves several times and then dried them for two days at 50 °C. The leaves were crushed with a rotter and then set aside to become a fine powder. Following impregnation with potassium hydroxide (KOH), the crushed leave were treated with KOH at the ratio of 1:1, then transferred to a crucible plate and heated at 1000 °C for one hour at 15 °C min–1. To remove impurities such as K, Mg, and so on, the sample was washed for 30 minutes with dilutes hydrochloric acid (HCl). As a final step after the HCl treatment, the product was washed with deionized water”.

Speaking of the nitrogen adsorption/desorption isotherm for the final composite material, we should apologize for the missing of such test, which may reflect the specific surface area that may offer more sites for reaction or catalysts.  The devices for this test are not available in our lab, we will try to conduct this test in the next related research work and evaluate the effect on the electrochemical performance. Thanks again for your valuable suggestion.

Reviewer’s Comment 3:  Unfortunately, the authors have not corrected the XRD pattern (Fig. 4a). It would also be proper to specify in the experimental part the parameters of the XPS study (cathode, power of the source, analysis features like a type of background, peak functions) and the wavelength for Raman.

Authors’ Reply: Thanks for the professional and helpful suggestions. We have revised the XRD patterns and the experimental parameters in the XPS and Raman tests in the revised manuscript as follows,

To characterize the active materials, the X-ray diffraction (XRD) spectra of the powder samples were measured using an X-ray diffractometer (Bruker D8 ADVANCE) with Cu Kα (λ = 0.15405 nm) radiation source and further analyzed using the X’Pert HighScore Plus software. The sample morphology was tested via transmission electron microscopy (TEM, JEM F200) and scanning electron microscopy (SEM, FEG 250). The thermal stability and the carbon content of the active materials were investigated using thermogravimetric and differential scanning calorimetry (TG-DSC, Mettler-Toledo). The crystal defects were evaluated via Raman spectrum using a Jobin LabRam high-resolution spectrometer, in a scan range of 500-4000 cm–1 with a laser source of 532 nm. A Thermo Fisher ESCALAB Xi+ spectrometer was used to analyze the C@Al2O3 structure and determine its chemical state. By using X-ray photoelectron spectroscopy (XPS, AXIS ultra DLD), using an Al Kα as the X-ray source. We measured the chemical composition of the active materials.

Reviewer’s Comment 4:  L. 224. Repeated use of XPS transcripts. Unfortunately, the authors ignored the question as to why there are no carbonyl groups in the activated carbon material.

Authors’ Reply: Thank you so much for pointing out this mistake. Repeated use of XPS transcript has been removed. The carbonyl groups (~286 to 287) in the XPS spectra of the composite material were added and analyzed in the revised manuscript.

Reviewer’s Comment 5:  L. 229. “These results indicate that the derived carbon is derived from flower waste leaves” – what does it mean? How XPS results can tell that?

Authors’ Reply: We should apologize for this incorrect expression about the XPS data analysis. To avoid the unnecessary misunderstanding about the XPS data, the incorrect statement was removed in the revised manuscript.

Reviewer’s Comment 6:  The interpretation of the XPS spectra is very poor. The authors give the spectrum of the O1s region, claiming that carbonyl groups are present there, but there is no such peak on the spectrum of the C1s region. In addition, in my opinion, the authors should be more thoroughly examined in the articles where carbon materials are studied using XPS. The point is that the authors incorrectly attributed the peaks on the O1s spectrum. The peak from carbonyl and carboxyl groups is in the region of lower binding energies compared to single C-O bonding.

Authors’ Reply: Thanks for the professional opinions on the XPS analysis. First, we should apologize for this mistake made in the XPS analysis. We have modified the XPS statements in the revised manuscript accordingly as follow,

The evaluate the surface element distribution, X-ray photoelectron spectroscopy (XPS) was utilized to capture a high-resolution spectrum of C@Al2O3. Figure 4c show the survey spectra of the desired elements of C@Al2O3 nanocomposite such as Al 2p, O1s and C 1s. Figure 4(d) presents the Al 2p peak due to the reaction between functional groups such as –NH2 and –OH groups, this 74.31 eV can be attributed to Al2O3 present on the surface of carbon layers [1, 2]. From the fit of O 1s spectra displayed in Figure 4(e), it can be found that there are three individual peaks centered at 531–532, 533–534, 534–536 eV, which correspond to C=O quinone type groups (O-I), C–OH phenol groups/C–O–C ether groups (O-II), and COOH carboxylic groups (O-III) [2,3]. For C@Al2O3 composite samples, the carbon–oxygen functional groups can improve the wettability of carbon materials further to enhance the chemical reaction on the electrode/electrolyte surface for obtaining extra electrochemical performance energy storage applications [4]. Figure 4(f) shows the XPS spectrum of the C 1s, which can be divided into major three peaks of C=C/C–C (284.6 eV), C=O (286–287 eV) and COOH (289–290 eV) [5, 6].

References:

 [1] N. Li, Z. Yi, N. Lin, Y. Qian, An Al2O3 coating layer on mesoporous Si nanospheres for stable solid electrolyte interphase and high-rate capacity for lithium ion batteries, Nanoscale, 11 (2019) 16781-16787.

[2] X. Han, Z. Zhang, H. Chen, R. You, G. Zheng, Q. Zhang, J. Wang, C. Li, S. Chen, Y. Yang, Double-shelled microscale porous Si anodes for stable lithium-ion batteries, Journal of Power Sources, 436 (2019) 226794.

[3] L. Lin, W. Lin, Y.X. Zhu, B.Y. Zhao, Y.C. Xie, G.Q. Jia, C. Li, Uniformly Carbon-Covered Alumina and Its Surface Characteristics, Langmuir, 21 (2005) 5040-5046.

[4] C. Long, L. Jiang, X. Wu, Y. Jiang, D. Yang, C. Wang, T. Wei, Z. Fan, Facile synthesis of functionalized porous carbon with three-dimensional interconnected pore structure for high volumetric performance supercapacitors, Carbon, 93 (2015) 412-420.

[5] X. Dong, H. Jin, R. Wang, J. Zhang, X. Feng, C. Yan, S. Chen, S. Wang, J. Wang, J. Lu, High Volumetric Capacitance, Ultralong Life Supercapacitors Enabled by Waxberry-Derived Hierarchical Porous Carbon Materials, Advanced Energy Materials, 8 (2018) 1702695.

[6] I.V. Plyuto, A.P. Shpak, J. Stoch, L.F. Sharanda, Y.V. Plyuto, I.V. Babich, M. Makkee, J.A. Moulijn, XPS characterisation of carbon-coated alumina support, Surface and Interface Analysis, 38 (2006) 917-921.

Reviewer 3 Report

The manuscript has been imporved.

Author Response

Thanks for the positive comments and valuable suggestion on this work.